# Using the Power of Junctional Adhesion Molecules Combined with the Target of CAR-T to Inhibit Cancer Proliferation, Metastasis and Eradicate Tumors

**DOI:** 10.3390/biomedicines10020381

**Published:** 2022-02-04

**Authors:** Christopher Mendoza, Dario Mizrachi

**Affiliations:** Department of Cell Biology and Physiology, College of Life Sciences, Brigham Young University, Provo, UT 84602, USA; cmendoz1@vols.utk.edu

**Keywords:** junctional adhesion molecule, chimeric antigen receptor, size-exclusion chromatography

## Abstract

Decades of evidence suggest that alterations in the adhesion properties of neoplastic cells endow them with an invasive and migratory phenotype. Tight junctions (TJs) are present in endothelial and epithelial cells. Tumors arise from such tissues, thus, the role of TJ proteins in the tumor microenvironment is highly relevant. In the TJ, junctional adhesion molecules (JAM) play a key role in assembly of the TJ and control of cell–cell adhesion. Reprogramming of immune cells using chimeric antigen receptors (CAR) to allow for target recognition and eradication of tumors is an FDA approved therapy. The best-studied CAR-T cells recognize CD19, a B-cell surface molecule. CD19 is not a unique marker for tumors, liquid or solid. To address this limitation, we developed a biologic containing three domains: (1) pH-low-insertion peptide (pHLIP), which recognizes the low pH of the cancer cells, leading to the insertion of the peptide into the plasma membrane. (2) An extracellular domain of JAM proteins that fosters cell–cell interactions. (3) CD19 to be targeted by CAR-T cells. Our modular design only targets cancer cells and when coupled with anti-CD19 CAR-T cells, it decreases proliferation and metastasis in at least two cancer cell lines.

## 1. Introduction

The tumor microenvironment (TME) is what surrounds a tumor, including the blood vessels, immune cells, fibroblasts, signaling molecules, and the extracellular matrix [1]. The tumor and its TME are closely related and interact constantly. Tumor cells achieve these interactions through cell-adhesion and recognition molecules, all members of the immunoglobulin superfamily (IgSF) [2,3,4]. Among the members of the IgSF are tight junction (TJ) components such as junctional adhesion molecules (JAMs) that act as gates and barriers to control the permeability of the paracellular space [5]. JAMs are an IgSF subfamily that contain four members: A, B, C, and 4. These components are also responsible for compartmentalization of the cellular environment and the separation of tissues [6]. JAM proteins form homotypic and heterotypic interaction among the same family members and may influence other members of the TJ [7,8]. Contrary to the effects of other TJ components [9], JAMs are responsible for increased proliferation when downregulated [10]. JAM-A upregulation has been associated with endothelial to mesenchymal transition (EMT) [11]. However, in glioblastoma cells, JAM-A may act as a tumor suppressor [12]. In our previous studies [7], we determined that JAM homo- and heterotypic interactions are of high binding affinity, resulting in increased cell-to-cell interactions. We have also determined that JAMs induced stronger cell adhesion than epithelial cadherin (E-CAD) [7,8]. Harnessing the function of JAMs in the TME may be of importance in translational solutions.

The IgSF proteins have been shown to play a role in cellular recognition [4]. Tumors often display unique proteins that are naturally targeted by immune cells surveilling the homeostatic landscape [13]. One of the most extensively used antigens for cancer immunotherapy is the B-cell-specific surface marker CD19 [14], used because of its expression in B-cell malignancies and lymphomas [15]. Therefore, an antigen recognizing CD19, known as anti-CD19 chimeric antigen receptor (CAR) T, was developed. It is fused to an intracellular signaling domain capable of activating T-cells to target and eradicate tumors [16]. These CAR-Ts have a surface receptor that works like an antigen recognition domain, which recognizes surface receptor targets such as CD19, leading to activation, cytokine secretion, and cellular proliferation, which in turn lead to tumor eradication. [16]. CD19-directed CAR-T cell therapy has been successful in treating several B-cell lineage malignancies, including B-cell non-Hodgkin lymphoma (NHL) [17,18]. CD19-directed CAR-T cell therapy is a powerful novel FDA-approved treatment that is currently being expanded to other immune cells [19] and to treat solid tumors [20]. Harnessing the power of CAR-T cell therapy is a promising future for cancer treatments.

Biologics are powerful treatments that can be made of sugars, proteins, DNA, or can be composed of whole cells or tissues. Human insulin was the first recombinant biopharmaceutical approved in the US in 1982 [21]. Protein-based therapeutics have been highly successful in the clinic and are recognized for their treatment potential [22]. Based on their pharmacological activity, they can be divided into the categories of: (a) replacing a deficient protein, (b) enhancing a pathway, (c) performing a novel function or activity, (d) interference, and (e) delivering other compounds or proteins [23]. They can be also be classified as non-covalent binding to their respective target, covalent bonding, and non-specific interactions with their respective targets [23]. New engineered proteins—including bispecific mAbs and multi-specific fusion proteins, antibody-drug conjugates, and proteins with optimized pharmacokinetics—are currently under development [23]. However, there are no conceptually new developments in protein-based biologics. There has been no protein engineering applied to new strategies in decreasing cancer metastasis. Computational designs are the theoretical approach to practical progress [24]. A paradigm change in the methodologies and understanding of mechanisms is needed to overcome major challenges like the complexity of biological systems, resistance to therapy, and access to targets.

Fusion proteins and fusion peptides as biologics have been described [25,26,27,28]. By joining different proteins that have different beneficial qualities, the potency, stability, and specificity of fusion proteins can be greatly enhanced compared with naturally occurring proteins [29]. PRS-343 is a bispecific fusion protein targeting HER2 and CD137, a costimulatory receptor on T-cells [30]. The PRS-343 architecture is derived from a trastuzumab variant and a CD137 specific anticalin [31]. Anticalins are engineered variants of tear lipocalin and neutrophil-gelatinase-associated lipocalin (NGAL), where loops are randomized by mutagenesis [32]. PRS-343 enables tumor-localized targeting of T-cells. This approach has the potential to provide a more localized activation of the immune system, resulting in higher efficacy and reduced peripheral toxicity [32]. Following this report, the authors initiated a phase I clinical trial with PRS-343 as a first-in-class molecule [32].

In this article, we propose a new type of protein-based biologic that can aid cancer treatment with CARs. We have designed a modular three-part biologic that creates new strategies pertaining to pharmacological activity, and activity or function. In this study, we generated a biologic that consists of three components: (1) A fusion peptide that detects cancer cells based on their lower pH. (2) JAM extracellular domain for binding to other TJ components in the TME that inhibits the metastasis of cancer cells. (3) A signaling target protein domain for CD19 that allows for the recognition of anti-CD19 CAR-T cells. Our biologic is modular in that the JAM or CD19 domains can be exchanged for tumor-specific proteins.

Inventions that aid CAR technology have been developed, such as the secretion of CD19-anti-Her2 bridging protein, that allow for T-cell cytotoxicity both in vitro and in vivo [33]. Other examples are the use of bispecific CAR-T to bring two cell types together such as cancer cells and T-cells [34,35], and the usage of donor stem cells or induced pluripotent stem cells to produce CAR specific treatments to derive natural killer cells, macrophages that can treat multiple myeloma [36,37,38]. These are very important contributions that target and treat cancer. In our study, we use a biologic to decrease cellular proliferation and metastasis.

Using JAMs as a part of our design allows for the formation of cell adhesion homotypic or heterotypic interactions, or both. We have reported that JAM-A binds to other members of the family but also coordinates the assembly of the TJ and the interplay with the adherens junction (AJ) [8]. The result of strengthening cell–cell interactions will subsequently result in decreasing metastasis in cancer cells. In order to introduce the extracellular domain of JAM proteins in the plasma membrane of tumor cells, we considered a peptide sequence known as pH-low-insertion peptide (pHLIP) that can recognize changes in pH and insert itself into the membrane of cancer cells [39,40]. The pHLIP inserts its C-terminus through a membrane under low pH conditions (6.0–6.5). This peptide has been used for the delivery of therapeutics [41]. In our design, pHLIP will allow the fusion biologic to target the cancer cell’s low pH and insert itself into the membrane, resulting in the extracellular domain of JAMs to be anchored to the cell surface.

The production of a biologic that inhibits metastasis of cancer cells would improve upon currently available targeted cancer treatment [42]. Inhibition of cancer cell proliferation and metastasis is important in controlling tumor growth, but it is also important to sensitize the tumor to therapeutics, decrease proliferation, and ultimately eradicate it. To accomplish this, we are using the extracellular domain of human CD19. The human CD19 antigen is a transmembrane protein belonging to the IgSF. CD19 is a biomarker for normal and neoplastic B cells, as well as follicular dendritic cells [43]. Indeed, CAR-T is a novel therapy that targets B cell malignancies based on their cell surface display of CD19 [44]. The addition of this extracellular CD19 domain to our biologic would represent an advantage that can be complimentary to existing CD19 CAR-T therapies for eradicating the tumor [45,46]. Our approach will increase cell–cell adhesion, preventing tumor growth and metastasis, while displaying CD19 on the surface of cancer cells. The combination of these three domains will not be restricted to blood malignancies alone but can benefit solid tumors as well.

The name of the biologic we have invented is CM19XA. The number 19 denotes CD19 while the letter A represents JAM-A soluble domain. The advantages of this biologic are that it contains three separate domains with distinct functions that target cancer cells, that can be used in any patient, regardless of their health. In this case, we are using this biologic to help the CAR-T cells target the tumor through the recognition of CD19, while decreasing metastasis. Our biologic is modular and CD19 may be exchanged for other cell surface biomarkers such as CD22, CD133, Her-2, EGFR, mesothelin, and others [47,48], and the JAM may be any of the four members of the subfamily, whichever is relevant to the tumor tissue of origin.

In order to use our biologic to decrease metastasis and to target these cancer cells for destruction by anti-CD19 CAR-T cells, we address the following questions: (1) Does CM19XA target cancer cells specifically? (2) Does CM19XA decrease metastasis by using the JAM components to establish and increase cell–cell interactions? (3) Does CD19 allow for the targeting of anti-CD19 CAR-T cells? (4) Does CM19XA work on other cancer cell lines?

## 2. Materials and Methods

### 2.1. Cloning, Protein Expression, and Purification

We synthesized the *E. coli* codon-optimized DNA sequence of CM19XA (Twist Bioscience, San Francisco, CA, USA). CM19XA was supplied by TWIST biosciences cloned in the expression vector pET28a, between restrictions sites NdeI and XhoI. A stop codon was introduced prior to the XhoI. The sequence upstream NdeI was the native sequence of pET28a which includes a 6xHis-tag sequence. pET28a CM19XA is preserved by transformation in the bacterial strain DH5α. Plasmid purification of a single bacterial colony was performed using the Zyppy Plasmid Miniprep Kit from Zymo Research. Sanger sequencing was performed by Genewiz (South Plainfield, NJ, USA) to determine whether the plasmid coding for CD19-JAMA-pHLIP was correct and that there were no mutations present. After the verification of the plasmid sequence, we transformed the plasmid into SHuffle T7 express bacterial cells (New England Biolabs, Ipswich, NY, USA) to compare which bacterial strain would give us the highest protein yield. Cells are grown to an OD_600_ of 1, followed by addition of 0.3 mM IPTG, and maintained at 16 °C for 18 h. The French Press method was used to lyse the transformed bacterial cells. The resuspended cells were loaded into the Thermo Spectronic French Pressure Cell Press Model FA-078. Lysis was performed at 1500–2000 psi using 30 mL of Wash Buffer consisting of 500 mM NaCl and 30 mM Tris and the lysate was collected in a 50 mL conical tube. Centrifugation was performed on the lysate for 30 min at 10,000 RPM with a F15-8 × 50cy rotor.

The supernatant was decanted into a 50 mL tube containing Ni-NTA Agarose beads from Prometheus (catalog no. 20-512) and incubated while rotating for 1 h at 4 °C. The column was washed with 100 mL of wash buffer containing 30 mM TRIS pH 7.5 and 500 mM NaCl, and 30 mM Imidazole pH 8.0. The supernatant was eluted for 3 min with 300 mM Imidazole, then concentrated using the Microsep Advance with 10 k Omega centrifugal device (reference no. MCP010C41) from Pall Corporation at 10,000 RPM for 10 min until a final volume of 2 mL was reached.

### 2.2. Size Exclusion Chromatography (SEC)

Size exclusion chromatography was performed using the NGC System (BioRad, Hercules, CA, USA). The column used was the ENrichTM SEC 65,010 mm × 300 mm, 24 mL, prepacked high-resolution SEC 650 column, with a size range of 5650 kDa (BioRad, Hercules, CA, USA). The protein peak was observed using the BioRad SEC software. The product peaks’ positions were compared relative to those of the size exclusion standards from BioRad (catalog No. 151-1901). Protein concentration was determined using the Nanodrop Onec from Thermo Scientific. The running buffer used was PBS, and proteins were also stored in PBS. Purification of JAM-A was performed as described in our previous publication [7].

### 2.3. SDS-PAGE Assay

Two µg of boiled MBP, JAM-A, and CM19XA were electrophoresed on 8% SDS-PAGE gel (BioRad). Gel staining was performed using standard protocols.

### 2.4. Tissue Culture and In Vitro Experiments with CAL27 and A549 Cells

Tongue squamous cell carcinoma cells (Cal27, ATCC CRL-2095) were obtained from American Type Culture Collection (ATCC, Manassas, VA, USA) and cultured according to the guidelines provided by the organization. RPMI, calcium-free with 10% FBS was used for all manipulations and the experimental set-up of CAL27 cells. A549 cells, epithelial cell lung carcinoma, were obtained from ATCC (catalog reference CCL-185).

### 2.5. Proliferation Assay

The first day of the proliferation assay consisted of 30,000 CAL27 cells seeded on 48-well plates. On the second day (at approximately 16 h), cells were treated with PBS or proteins at a final protein concentration of 1 μM (MBP-JAMA or CM19XA). After 24 h, proliferation assays were performed using ATPlite Luminescence Assay System (product number 6016943, PerkinElmer, American Fork, UT, USA) following the manufacturer’s instructions. After 72 h of mock or anti-CD19 CAR-T killing assay for both CAL27 and A549 cells, we performed the ATPlite Luminescence Assay [49,50].

### 2.6. Wound Healing Assay

The wound healing assay was performed as follows: on the first day, 15,000 CAL27 cells were seeded on each chamber of the 2-well silicone insert (IBIDI, Gräfelfing, Germany) separated by a silicone gap of 500 ± 100 µm. After 24 h, proteins were added at a final concentration of 1 µM, JAM-A or CM19XA. Cells were incubated with the treatments for 2 h at 37 °C. Following the treatment, the silicon insert was removed. The wells were rinsed once with PBS and then each well was filled with DMEM F-12 media with 10% FBS (Genesee Scientific, El Cajon, CA, USA). The closure of the gap (500 µm ± 100 µm at time zero) was evaluated 16 h post treatment using an Olympus IX70 microscope (Olympus Life Science, Waltham, MA, USA). Images were analyzed using cellSens Entry Microscopy Imaging Software by Olympus Life Science. The distance of the gaps was then quantified using ImageJ [51]. Data analyzed using FastTrack AI (IBIDI, Gräfelfing, Germany) [52].

### 2.7. Real-Time Cell Invasion

Real-time cell invasion was determined after the various treatments. The xCEL-Ligence RTCA cell monitoring system was used to quantify real-time invasion of cells per the protocol suggested by the manufacturer (ACEA Biosciences, Blue Springs, MO, USA). The invasion was performed in 16-well CIM plates (*n* = 10 groups per treatment, ACEA Biosciences, Blue Springs, MO, USA). The tops of the wells were coated using a 1:40 Matrigel concentration (Fisher Scientific, Pittsburg, PA, USA). Then, a concentration of 20,000 CAL27 cells was used with 100 μL of 2% FBS RPMI, with and without proteins. The bottom chamber wells were treated with 160 μL of 10% FBS RPMI. Cells were placed in the xCeLLingence RTCA instrument, where the invasion readings were taken 4 times an hour for 24 h.

### 2.8. CAR-T Cell Killing Assay

On the first day, 5000 CAL27 or A549 or HUVEC cells/well (catalog C0035C, Thermo Fisher Scientific, Waltham, MA, USA) were plated on a 96-well plate. On day 2, cells were incubated for 3 h under the following conditions: 14 wells had no treatment, which was used as a control. In 16 of the wells, 1 µM soluble JAM-A protein was introduced. In 14 of the wells, 1 µM of CM19XA was introduced. On day 2, both mock CAR-T and anti-CD19 CAR-T were dispensed to half of the wells in each treatment. The killing assay was allowed to continue for a total of 72 h.

### 2.9. Computer Models of the Biologic

Protein models were produced using UCSF Chimera v. 1.15 package from the Resource for Biocomputing, Visualization, and Informatics at the University of California, San Francisco (supported by NIH P41 RR-01081) [53,54].

### 2.10. Statistical Analysis

Student’s *t*-test was performed using GraphPad Prism version 8.0 to generate the graphs to compare control vs. JAM-A and CM19XA.

## 3. Results and Discussion

### 3.1. Expression and Purification of CM19XA in E. coli

Extracellular domains of JAM-A were expressed as a fusion with CD19 and pHLIP containing an N-terminal 6x-HIS tag to allow for the use of Nickel NTA purification strategy (Appendix A). The resulting biologic protein will be called CM19XA after the inventor of the biologic, Christopher Mendoza. CM19XA was modeled using UCSF Chimera [53,54] to determine the folding of the protein (Figure 1A). The model shows the CD19 (coral), with a Gly-Ser linker (gray) that connects JAM-A and the pHLIP peptide (yellow). After modeling, CM19XA was subcloned in a Kanamycin-resistant pET28a plasmid (Figure 1B). Since two domains of CM19XA (CD19 and JAM-A) require proper disulfide formation to allow for the correct folding and function, we used the SHuffle T7 Express bacterial strain [55,56]. This bacterial strain allows for the cytosolic expression of target proteins, while enabling proper disulfide bond formation and high protein yield [55]. The plasmids hosting CM19XA, or JAM-A were transformed into SHuffle T7 bacterial cells and grown at 37 °C in LB containing kanamycin (required by pET28a) and spectinomycin (required by SHuffle cells). Ni-NTA resin affinity chromatography was followed by Size Exclusion Chromatography (SEC). The results of the purification of CM19XA by SEC showed dimers (Figure 1C). The formation of dimers is consistent with our previous publication where we observed dimer formation with JAM-A [7]. The protein size was determined to be about 61.5 kDa (Figure 1D), as expected (Appendix A). JAM-A has a size of 70 kDa (Figure 1D) because this protein is fused with MBP, as reported in our previous studies [7].

### 3.2. Proliferation Assay

To determine whether CM19XA targets cancer cells, we used the cell line CAL27 in a proliferation assay. In our laboratory we have identified that the membrane composition of the TJ is simple, consisting of claudin-1, JAM-A, and occludin. In the literature, the downregulation of JAM-A through siRNA leads to cell proliferation [10]. We have confirmed this result and expanded the report by comparing the effects of the soluble extracellular domain of JAM-A and CM19XA in CAL27 cells. The comparison of these two proteins resulted in opposite CAL27 cell behavior in the proliferation assay (Figure 2) that equates ATP production to cell proliferation. JAM-A increases the proliferation of the CAL27 cells with a ratio value of 1.239 compared to 1.000 obtained in the control (no proteins added). This means that JAM-A is increasing the proliferation of CAL27 cells, perhaps by disrupting *trans*-interactions and fostering *cis*-binding to native JAM-A. This result agrees with the literature [10]. In contrast, CM19XA decreased the proliferation of CAL27 cells, 0.783-fold of the control. This means that CM19XA is functional by first inserting the pHLIP peptide into the membrane of the CAL27 cells, and second, the JAM-A domain establishes cell-to-cell interactions that lead to decreased CAL27 proliferation.

The proliferation assay of CAL27 cells shows that there is a decrease in the proliferation with the addition of CM19XA. The proliferation assay was different for the introduction of JAM-A, where there was an increase of proliferation as compared to no effect with the control. The data above shows that soluble JAM-A decreases cell–cell adhesion, leading to an increase in cellular proliferation. The biologic CM19XA increased cell-to-cell adhesion by increasing JAM protein binding to other TJ proteins either by homotypic or heterotypic interactions [7]. This could be due to JAM-A not inserting itself into the membrane and binding to other JAMs (-A, -B, -C, and 4) in *cis* that cause an interruption in the cell-to-cell interactions meaning that there is a decrease in TJ *trans* interactions. In the case of CM19XA, the decrease in proliferation is caused by the pH sensitive region of the biologic inserting itself into the membrane of the cancer cells, which allows the JAM-A portion of the biologic to reinforce the binding of native JAM proteins in *trans* leading to greater cell-to-cell interactions. Therefore, the phenotype observed seems consistent with decreased proliferation because CM19XA increases binding of TJ components such as JAMs.

### 3.3. Wound Healing Assay

In order to validate the ATP proliferation experiments where JAM-A increases CAL27 proliferation and CM19XA decreased CAL27 cell proliferation, we performed a wound healing assay. This assay would allow us qualitatively and quantitatively determine whether the JAM-A portion of CM19XA would increase cell–cell interactions, and as a result, decrease cell migration, resulting in a larger gap. Cell migration is important in many physiological processes that are heavily regulated. Wound healing and cancer cell migration assays are widely used for the understanding of cues that can increase or decrease cell migration. Thus, we decided to use CM19XA with CAL27 cells to determine the effect this biologic had on cell migration by performing wound healing assays.

The wound healing experiments shows the effects of our biologic, CM19XA first qualitatively (Figure 3A–C). We determined that soluble JAM-A increased cell migration, and as a result the gap created at the beginning of the experiment (500 µm) was decreased to an average of 88 µm determined from four independent experiments. Without addition of proteins (control), the gap closure was in average of 153 µm. This validated the ATP proliferation results shown in Figure 2, where JAM-A increased CAL27 cell proliferation. The drastic result was observed with CM19XA where the gap was greater than the control and JAM-A, at 285 µm in average. This demonstrates that CM19XA is functional, and the JAM-A portion of the protein is able to establish cell–cell interactions, decreasing cell migration (Figure 3) and proliferation (Figure 2). The analysis of the data as the mean ± SD is presented in Figure 3D.

### 3.4. Cell Invasion Assay

Based on the results of the wound healing assay, we decided to determine the effect of CM19XA on cell invasion. Cell invasion recordings were observed to be low for the control containing only 10% FBS. JAM-A resulted in an increase of cell invasion (Figure 4A), which could be due to how the protein is binding in *cis* to other JAMs, decreasing cell–cell interactions and promoting proliferation. JAM-A increases cell invasion by about 8-fold compared to the control, which is consistent with the increase in proliferation (Figure 2) and decreased gap in wound healing (Figure 3). CM19XA de-creased invasion by 25% compared to the control (Figure 4B). The decrease in cancer cell invasion by CM19XA is consistent with the decrease in proliferation (Figure 2), and an increase of gap formation in the wound healing assay (Figure 3), meaning that this biologic is binding in trans to other JAM proteins that decrease cellular proliferation, increase TJ formation, and as a result, decrease cell invasion (metastasis).

Cell invasion recordings were observed to be low for the control containing only 10% FBS. JAM-A resulted in an increase of cell invasion (Figure 4A), which could be due to how the protein is binding in *cis* to other JAMs, decreasing cell–cell interactions and promoting proliferation. JAM-A increases cell invasion by about 8-fold compared to the control, which is consistent with the increase in proliferation (Figure 2) and decreased gap in wound healing (Figure 3). CM19XA decreased invasion by 25% compared to the control (Figure 4B). The decrease in cancer cell invasion by CM19XA is consistent with the decrease in proliferation (Figure 2), and an increase of gap formation in the wound healing assay (Figure 3), meaning that this biologic is binding in *trans* to other JAM proteins that decrease cellular proliferation, increase TJ formation, and as a result, decrease cell invasion (metastasis).

### 3.5. CM19XA Only Targets Cancer Cells

In order to determine whether CM19XA only targeted cancer cells, we used the non-cancerous HUVEC cell line. We determined that addition of soluble JAM-A protein increased cellular proliferation (Figure 5A) as seen in the previous cell lines. However, CM19XA did not increase the killing of HUVEC cells (Figure 5A). This is due to the pHLIP peptide portion of the biologic not inserting itself into the membrane of the cells. Therefore, anti-CD19 CAR-T cells are not recognizing the HUVEC cells or CM19XA, and, as expected, no killing occurs. This shows that CM19XA is cancer specific.

To determine whether CM19XA targets other cancer cells, we repeated the cytotoxicity experiments with A549 lung cancer cells. The cytotoxicity cell assay in Figure 6B demonstrates that CAR-T is able to recognize the extracellular portion of CM19XA and kill the target CAL27 cells (Figure 6B). The same effect seen with CAL27 cells was seen in A549 cells in proliferation assays after 72-h post CAR-T killing (Figure 5C). JAM-A in CAL27 cells in the mock experiment showed an ATP fold increase of 1.39, similar to 1.23 in A549, suggesting an increase in proliferation. Similar results were seen in anti-CD19 with JAM-A for CAL27 cells: the ATP fold increase was 1.24 and for A549 the result was 1.18 suggesting that in both cases, JAM-A increases cellular proliferation compared to both the mock control and the CD19 control (Figure 6). When comparing to the CM19XA mock we see that the ATP fold increase for Cal27 cells is 1.01 and 1.03 for A549 which is similar to 1.00 as seen mock controls (Figure 5). This means that the mock CAR-T cell lines do not kill the target cell lines since they are not able to recognize the extracellular CD19 domain of CM19XA. However, when anti-CD19 CAR-T cells were used, we see that there is decrease in the proliferation of 0.72 fold for CAL27 cells and 0.74 fold for A549 (Figure 5), suggesting that the CM19XA is able to work in both cell lines by the following: (1) Inserting itself using the pHLIP peptide, (2) Increasing tight junction formation by JAM-A binding in *trans* to other JAM proteins that increase cell-to-cell interaction, (3) CD19 acting as a recognition signal for anti-CD19 CAR-T cells to recognize and kill the targeted cancer cells.

### 3.6. Cytotoxicity Assay

To determine whether the CD19 portion of CM19XA worked as a signal to allow for CAR-T cells to recognize and kill cancer cells, we performed cytotoxicity assays. We used cultured cells as a model system to determine the functionality of extracellular CD19 as a target signal for anti-CD19 CAR-T cells. Cells were observed to grow without problems for mock CAR-T experiments using a CAR-T cell that did not recognize the CD19 portion of CM19XA as expected. There was no effect on the killing of CAL27 cells with the mock CAR-T experiments using control (no protein), JAM-A, and CM19XA (Figure 6A). In contrast, when using the anti-CD19 CAR-T cells that recognize extracellular CD19, differences were observed. There was no killing in the control (no protein), or JAM-A condition, but there was an increase in killing when CM19XA was used (Figure 6B). The increase of killing of anti-CD19 CAR-T cells targeting CM19XA was observed in the decrease of proliferation of CAL27 cells (Figure 5B).

### 3.7. Proposed Mode of Action of CM19XA

Based on our design and experimental data, we present our biologic, CM19XA which will identify and target cancers according to the decreased pH of the membrane of cancer cells (Figure 7). The pHLIP peptide will proceed to insert into the membrane displaying the two soluble domains, CD19 and JAM-A. JAM-A will increase cell-to-cell interactions, decreasing proliferation. Over the last decade, the key role of the TJ in tumor progression and metastasis has been established [57]. In addition to its role in the control of paracellular diffusion, the TJ has a vital role in maintaining cell-to-cell adhesion and tissue integrity. Thus, CM19XA, or any of its derivatives where the JAM domain is replaced by JAM-B, -C, or 4, strengthens the TJ and cell-to-cell interactions (Figure 8). The modularity of our biologic can address the differences in TJ composition due to tissue-specific expression of its membrane components. CD19 can be used as a recognition signal for anti-CD19 CAR-T cells.

The identification of pro- and anti-cancer roles among TJs such as claudins has been puzzling [58]. Similarly, the role of JAM-A and JAM-C in the progression of malignant neoplasm has been described to have a number of contradicting phenotypes. The role of JAM proteins in cancer is complex. JAMs function by interacting with other proteins via several mechanisms: direct cell–cell interaction on adjacent cells, stabilization of adjacent cell surface receptors on the same cell, and interactions between JAM and cell surface receptors expressed on adjacent cells. The diverse interactions contribute to both the pro- and antitumorigenic functions of JAM [59]. This paradigm can also be observed in a study that presents evidence that JAM-A knockdown accelerates the proliferation and migration of human keratinocytes [10]. On the other hand, Solimando et al. studied the role of JAM-A in multiple myeloma (MM) [60]. In vitro JAM-A inhibition impaired MM migration, while in vivo treatment with an anti-JAM-A monoclonal antibody impaired tumor progression [61]. These results could correspond to JAM-A interactions and effects within the same cell or to a signal transduction that is not fully understood [5,6]. The importance of mechanical transduction [62] from cellular junctions, both TJ and AJ, is poorly understood but the work by Solimando [60,61] and others highlight the need for further study of this phenomenon [59]. Additionally, a report that JAM-A functions in a tumor-suppressive role by increasing apoptosis and suppressing proliferation in colorectal adenocarcinoma revealed that loss of JAM-A expression increased intestinal epithelial cell proliferation [63]. The relevance of this paradigm may simply indicate that regulation of JAM-A expression in the context of cell proliferation may be tissue- and cell-specific.

Considering that JAM-A is a key player that coordinates TJ and AJ’s interplay [8], understanding its function in tumorigenesis and metastasis is of high relevance. While assessing invasive breast cancer, data was presented showing that cell lines with the lowest migratory capacity (T47D and MCF-7 cells) express higher levels of JAM-A relative to more migratory lines (MDA-MB-231 cells). Ectopic expression of JAM-A in these highly metastatic cells diminished both cell migration and invasion [64]. On the contrary, silencing of JAM-A expression enhanced the invasiveness of the less migratory lines [64]. Nevertheless, evidence for the opposite phenotypes can also be found in the literature. Functional inhibition of JAM-A protein activity inhibits the adhesion and trans-endothelial migration of breast cancer cells [65]. Human nasopharyngeal cancer cells exhibit increased JAM-A levels, which leads to increased endothelial-to-mesenchymal transition [11]. In lung adenocarcinoma, the suppression of JAM-A expression by siRNA inhibited cellular motility and invasiveness, while JAM-A inhibition caused a decrease in colony-forming capability in vitro and an inhibition of tumorigenicity in vivo [66].

As a final consideration, CM19XA lacks the capability of intracellular signaling. This could be a reason why the results we observed deviate from what could be expected according to the previous discussion. CM19XA is capable of carrying out two functions once inserted in the membrane through pHLIP. First, CD19 attracts CAR-T cells; second, JAM-A interacts with other TJ membrane proteins (*trans* and *cis* interactions) and also exhibits self-interaction; both will result in *cis* and *trans* interactions. From this point of view, CM19XA can be examined for its role as an adhesion molecule rather than its signal transduction leading to tumor related phenotypes. If our hypothesis is correct, then the idea that regulation of JAM-A expression in the context of cell proliferation may be tissue- and cell-specific will not apply. We imagine that as an adhesion promoting agent CM19XA will be tissue independent. Tissue specificity may require utilizing a different biology (Figure 8) among the modular designs we have prepared. Homeostasis in healthy tissues strongly relies on cell-to-cell adhesion and cell-to-extracellular matrix interactions. Despite many studies describing the relationships between malignant transformation, metastasis, and cellular adhesion processes, many questions remain [67]. Cadherins and integrins are among the most studied classes of adhesion receptors. Integrins play a key role in single-cell migration, which requires the complete loss of AJs mediated by epithelial cadherin (E-CAD). In malignant transformation in the epithelium, cells lose their dependence on integrin-mediated interactions with the extracellular matrix. During this process, AJs and E-CAD are lost along cell–cell interactions [68]. On the other hand, loss of E-CAD inhibits CD103 antitumor activity, reducing checkpoint blockade responsiveness in melanoma [69]. Restoring E-CAD could be a potential approach for cancer therapy. Multiple natural compounds have been shown to possess anti-tumor activities through the regulation of key molecules in signaling pathways, including restoring E-CAD cell–cell adhesion [70]. Considering the previous argument, we suggest that CM19XA acting mostly as a cell-adhesion enhancer is capable of decreasing metastasis. Considering that JAM-A may interact with E-CAD [8], then a possibility for signal transduction via CM19XA:E-CAD interactions cannot be dismissed, and should be studied in the future.

The modularity of our biologic can address the differences in TJ composition due to tissue-specific expression of its membrane components. CD19 can be used as a recognition signal for anti-CD19 CAR-T cells. In traditional anti-CD19 CAR-T therapies, the cell targets the naturally displayed CD19 of B cell malignancies. The modularity of our biologic will enable the selection of any surface biomarker desired based on the tumor type.

## 4. Conclusions

We have designed and tested CM19XA, a three-domain biologic. We have presented evidence that our biologic inserts itself into cancer cells using its pHLIP peptide. The second domain of the biologic, JAM-A, increases cell-to-cell interactions that in turn decrease proliferation and may prevent tumor cells from leaving their niche, inhibiting metastasis. The third domain of the biologic, CD19, is recognized by anti-CD19 CAR-T cells, allowing for targeted cancer cell killing. Our biologic produced similar results in two cell lines, CAL27 and A549 and had no effect on the non-cancer cell line HUVEC, showing that it is cancer specific. This suggests that CM19XA may be used as a therapeutic that will recognize multiple cancer cell lines. CM19XA’s adhesive properties provide increased cell-to-cell interactions through its JAM domain, and cellular recognition of immune cells through the CD19 domain, resulting in cancer cell killing. We suggest that CM19XA is a new classification of protein-based biologic that pairs with current CAR therapies to recognize cancer cells, increases cell-to-cell interactions that lead to a decrease in proliferation and metastasis, and increases cancer cell killing.

CAR-T is produced from a patient’s blood, where the gene for a single receptor is inserted with the purpose of attacking a specific cancer cell. These genetically engineered T-cells are then re-introduced into the patient. Looking forward, the CM19XA design will advance the treatment of cancer by serving as an additional tumor specific mechanism. This biologic can be manufactured at large scale and can be used to target CD19, CD38, or other specific tumor targets using the corresponding CAR-T cell design. Depending on the type of tumor cell to be targeted, the JAM proteins (A, B, C, and 4) can be interchanged. Future work will include in vivo experimentation and characterization of potency, stability, and specificity. Analysis of CM19XA and its derivatives will involve cytokine release syndrome (CRS) grading and other safety measurements as the research progresses.

## 5. Patents

Patent resulted from this work, 2021. Provisional patent EFS ID: 44186851. Patent sponsored by Brigham Young University, Tech Transfer Office. Director Michael Alder, mike_alder@byu.edu.

## Figures and Tables

**Figure 1 biomedicines-10-00381-f001:**
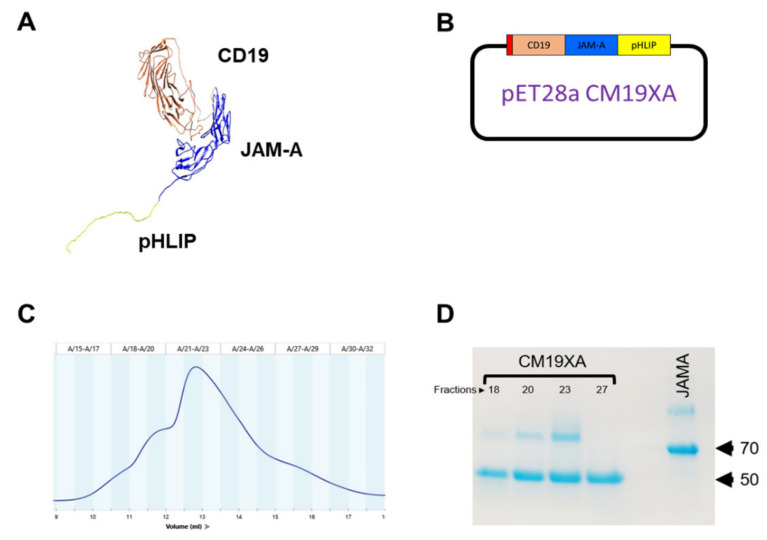
CM19XA modeling and purification. (**A**) Computer model of CM19XA. CM19XA consists of CD19 (coral), GS linker (Gray), JAM-A (blue), and pHLIP peptide (yellow). (**B**) Plasmid (pET28a) hosting CM19XA, contains a 6xHis tag, N-terminal to CD19. (**C**) Size-exclusion chromatogram of CM19XA purification. (**D**) Coomassie stain of CM19XA purified fractions and JAM-A as a control.

**Figure 2 biomedicines-10-00381-f002:**
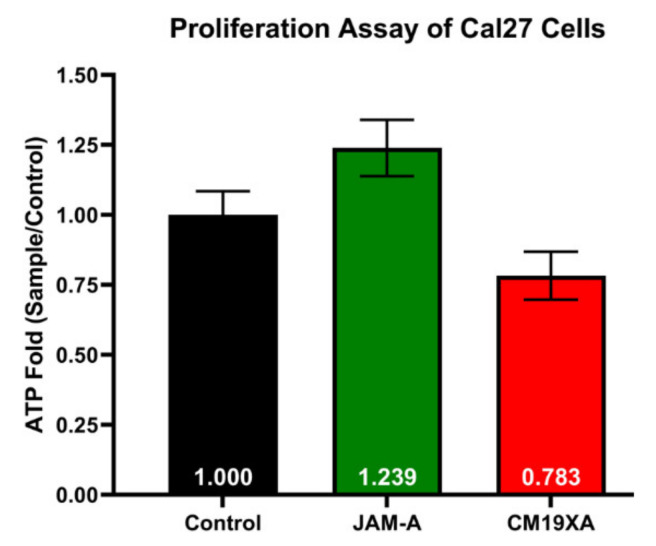
CM19XA decreases Cal27 cell proliferation. Data are expressed as fold of CAL27 proliferation without treatment (control). JAM-A compared to the control increases proliferation, while CM19XA caused a decrease in proliferation compared to the control and JAM-A. This decrease in proliferation is due to the pHLIP portion of CM19XA inserting itself into the membrane, allowing for the JAM-A portion to bind to other JAM proteins in *trans.* As a result, the binding of JAM-A of CM19XA in *trans* allows for an increase in cell-to-cell interaction. Statistical analysis using Student’s *t*-test was performed.

**Figure 3 biomedicines-10-00381-f003:**
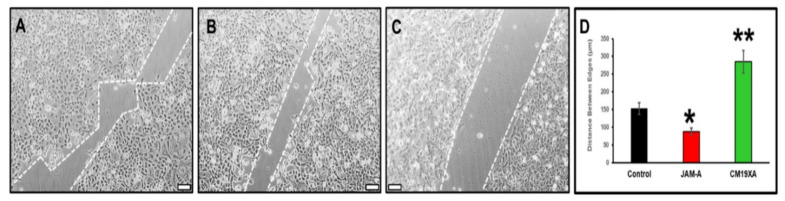
Wound healing assay. The movement of CAL27 cells 24 h after removing the silicon inserts that separate the cells. The gap size at time 0 h is 500 µm. Culture conditions: (**A**) control, no addition of proteins, (**B**) 1 µM JAM-A, (**C**) 1 µM CM19XA. For each condition, at least 900 individual cells were tracked using FastTrack Artificial Intelligence (AI) automated analysis system, an AI-based vision system from MetaVi Labs/Ibidi (cat no. 32200-3; IBIDI, Gräfelfing, Germany). (**D**) Statistical analysis of the movement of CAL27 cells. The graph reports the average distance between edges of CAL27 cells. The data from four independent wound healing assay experiments are shown as the mean ± SD. The statistics show that when comparing control versus JAM-A (*) results in a *p* < 0.04. When comparing the control versus CM19XA we get (**) a *p*-value of *p* < 0.01. These results show that the bigger gap from CM19XA cells is from an increase in TJ formation, which leads to a decrease in cell proliferation and migration. However, with JAM-A there was a decrease of the gap, showing that these cells are proliferating and migrating due to a decrease in TJ formation. These results show that CM19XA decreases the migration from the cancer cells, which is reflective of the decrease in proliferation. Scale bar in panels (**A**–**C**) represents 100 µm.

**Figure 4 biomedicines-10-00381-f004:**
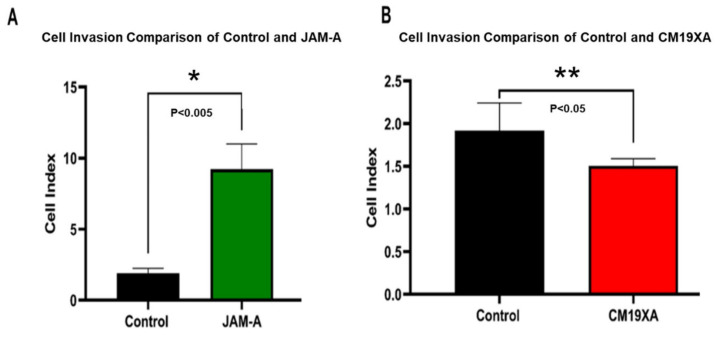
Cell invasion assay of CAL27 cells. (**A**) Comparison of Cell Index in the absence of exogenous proteins (control) or 1 µM JAM-A. JAM-A triggers an increase of almost 8-fold invasion. (**B**) Comparison of Cell Index in the absence of exogenous proteins (control) or 1 µM CM19XA. CM19XA decrease the rate of invasion by 25%. Statistical analysis was performed using Student’s *t*-test. Panel A, (*) *p* < 0.005; Panel B, (**) *p* < 0.05.

**Figure 5 biomedicines-10-00381-f005:**
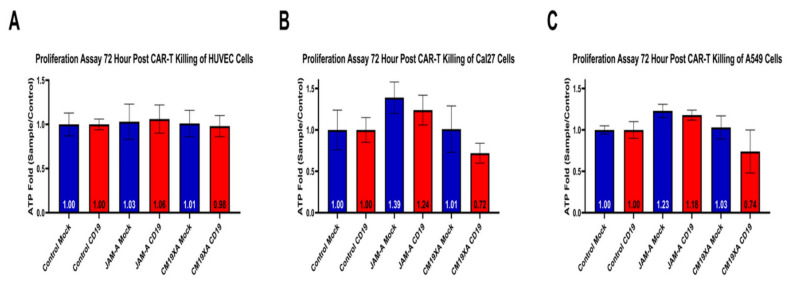
Proliferation Assay after 72-h post CAR-T cytotoxic assay of HUVEC, CAL27, or A549 cancer cells. (**A**) As expected, CM19XA did not increase killing of HUVEC cells. This is because the pHLIP portion of the protein did not insert itself into the membrane of the noncancerous HUVEC cells. (**B**) Control resulted in the same results where there was no change in the proliferation, as expected. JAM-A resulted in an increase of proliferation in both mock and anti-CD19 CAR-T, validating the results from the wound healing assay and the other proliferation assays. CM19XA mock had similar results in proliferation to that of the control. CM19XA treated with anti-CD19 CAR-T cells had a decrease in proliferation, showing that there is an increase in killing. This means that the CD19 component of the CM19XA biologic can be detected by anti-CD19 CAR-T cells. (**C**) Control resulted in the same results where there was no change, similar to that of CAL27 cells. JAM-A resulted in an increase in proliferation that is similar to the results with the CAL27 cells. CM19XA mock had similar results as seen with CAL27 cells, where there was no killing of CAR-T cells, resulting in similar proliferation as the control. A549 cells CM19XA treated with anti-CD19 CAR-T showed similar results from CAL27 cells, where there was an increase in killing that reflects the decrease in proliferation. Overall, this experiment shows that CM19XA can be recognized by anti-CD19 CAR-T cells via the extracellular component of the biologic, increase cancer cell killing, and decrease proliferation. Statistical analysis was performed using Student’s *t*-test, not statistical difference for panel A (*p* < 0.1). Differences were observed with statistical significance for CAL27 (*p* < 0.01) and A549 (*p* < 0.03).

**Figure 6 biomedicines-10-00381-f006:**
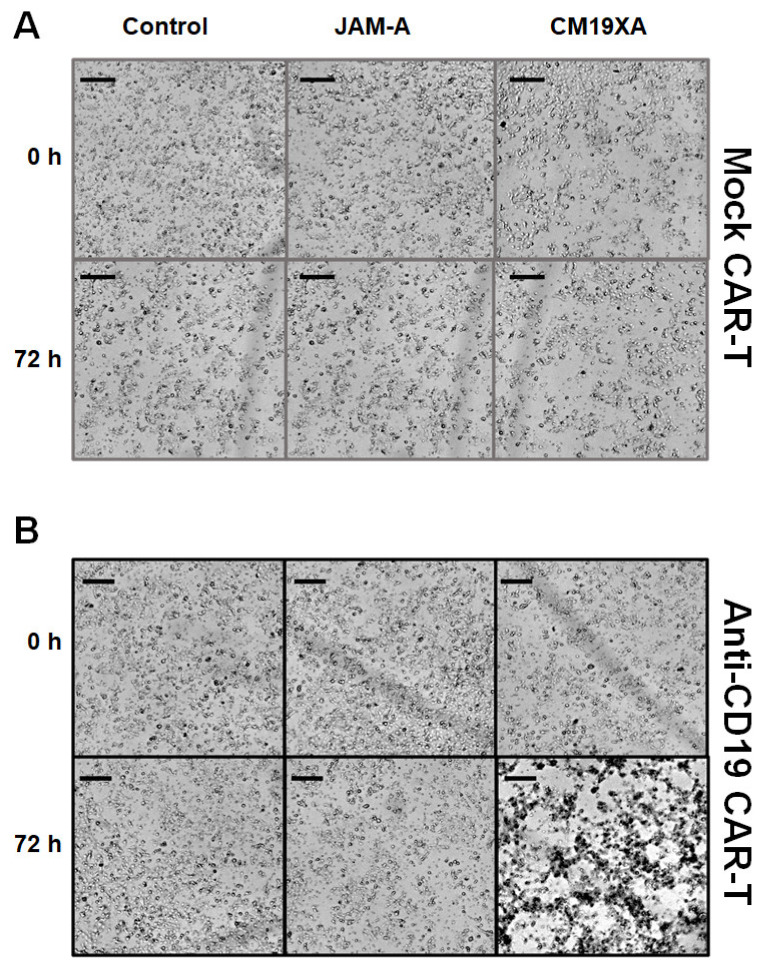
Cytotoxicity assay. Images of CAL27 cells at 0 and 72 h post protein treatment. Cells were incubated with either mock or CD19 CAR-T with a ratio of 1:5 (CAL27: CAR-T). (**A**) There was no killing observed in mock CAR-T (CAR-T not able to recognize CD19) with the control (no protein), JAM-A, and CM19XA. (**B**) Anti-CD19 CAR-T cells had no killing from 0 to 72 h in the presence of the control or JAM-A. There was an increase in target killing of CAL27 cells in the presence of CM19XA, suggesting that the extracellular CD19 in the biologic can work as a recognition site for anti-CD19 CAR-T cells. Scale bar is 100 µm.

**Figure 7 biomedicines-10-00381-f007:**
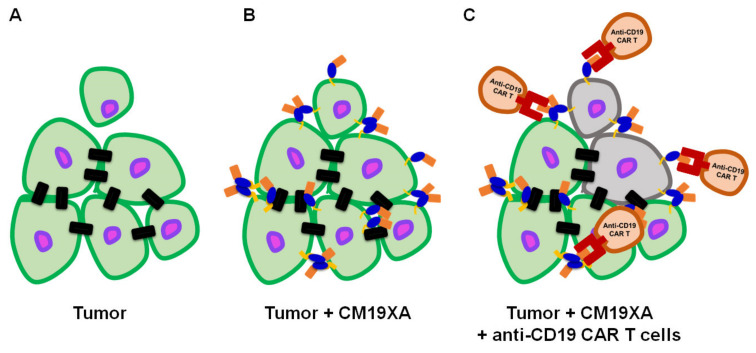
CM19XA mechanism of action. (**A**) Tumor formed by cells (green with purple nucleus) establishing cell–cell interactions (black rectangles). One cell poses a risk for metastasis after losing cell attachments to the main tumor. (**B**) pHLIP (yellow) inserts itself into the membrane of cancer cells. JAM-A (blue) binds to other JAM proteins in the existing cell–cell interactions in *trans* allowing for homotypic or heterotypic interactions to occur. Additionally, CM19XA may establish de novo cell–cell interactions by creating *trans* homotypic interactions. CM19XA will restrain weakly interacting cells from metastasizing, decreasing proliferation and metastasis. (**C**) Anti-CD19 CAR-T cells (brown) will recognize the tumor cells displaying CD19 (coral) and proceed to eradicate the tumor (gray).

**Figure 8 biomedicines-10-00381-f008:**
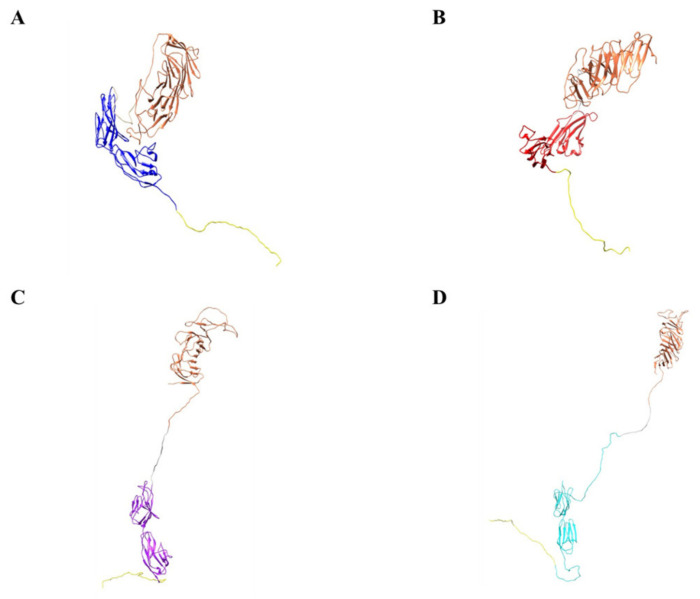
Computational models demonstrate modularity of metastasis-inhibiting protein. (**A**) CD19 (coral), GS linker (gray), JAM-A (blue), and pHLIP peptide (yellow). (**B**) CD19 (coral), GS linker (gray), JAM-B (red), and pHLIP peptide (yellow). (**C**) CD19 (coral), GS linker (gray), JAM-C (purple), and pHLIP peptide (yellow). (**D**) CD19 (coral), GS linker (gray), JAM 4 (cyan), and pHLIP peptide (yellow).

## Data Availability

PDB models are available upon request.

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
