# Peer review of "Using the Power of Junctional Adhesion Molecules Combined with the Target of CAR-T to Inhibit Cancer Proliferation, Metastasis and Eradicate Tumors"

_biomedicines, 2022, doi:10.3390/biomedicines10020381_

Round 1

Reviewer 1 Report

First of all, I would like to congratulate the authors for their work in bringing to the forefront knowledge in the field of cancer therapy. The research has much potential for the realisation of the results through their application in the routine treatment of patients.

However, I would like to make a few suggestions.

  1. The introduction is far too broad. There are many details that should be moved to the discussion section.
  2. The results and discussion sections are merged, which is not wrong, but the importance of your results should be highlighted by creating separate sections. Moving the information from the introduction, which is necessary, will further overshadow its value.
  3. Explain more fully the fusion phenomenon and elaborate on fusion peptides.
  4. “The production of a biologic that inhibits metastasis of cancer cells would improve upon currently available targeted cancer treatment [34].” “The addition of this extracellular CD19 domain to our biologicwould represent an advantage that can be complimentary to existing CD19 CART therapies for eradicating the tumor [37,38].” Here you state about the biologic you produce. Why put the references if you are talking about your product?

Author Response

Thank you.

Reviewer 2 Report

Christopher Mendoza and Dario Mizrachi try to claim an effect of CAR-T to inhibit cancer proliferation and metastasis. The topic is of interest, but the data are preliminary.

Major points

  1. The authors employed squamous cell carcinoma cells (ATCC CRL-2095) and CAL27 cells. Since they claim to halt EMT, and JAM-A overexpression is related to disease progression in diffuse large B-cell lymphoma and downregulated by lenalidomide, another hematological cell model should also be employed (PMID: 28785100)
  2. A wound healing assay is not sufficient to support a robust statement regarding the potential of their approach of halting cancer cells invasion.
  3. Moreover, did the authors normalize the scratch wound assay for cell number, proliferative index, viability?
  4. The image quality for the scratch is suboptimal and a robust quantification would be needed to make the results trustworthy and reproducible (There are open-source programs such as imageJ to analyze images of in vitro scratch wound healing assays, but these tools require manual tuning of various parameters, which is time-consuming and limits image throughput. For that reason, an optimized plugin for imageJ to automatically recognize the wound healing size, correct the average wound width by considering its inclination, and quantify other important parameters such as: area, wound area fraction, average wound width, and width deviation of the wound images obtained from a scratch/ wound healing assay has been developed. This would make the experiment consistent.
  5. Student’s t-test was performed: were the data normally distributed? Could a parametric test be employed?
  6. Additional and alternative approaches should corroborate and replicate results regarding invasiveness, migration, proliferation and apoptosis
  7. There is a lack of biological explanation from the mechanism standpoint It would be of relevance to perform a wide-transcriptome profiling of the JAM-A-neutralized-cells; and have the most significantly modulated genes to be functionally studied. This would provide novel insights into the molecular mechanisms underlying JAM-A-driven effect on cancer functions.
  8. The mechanistic impact of biological targeting JAM-A in normal cells versus cancer cell lines on a cell growth and migration needs to be investigated as well; for a high impact journal it is not enough to hypothesize on specific functions based on array data and few experiment.
  9. Figure 6 quality is not satisfactory: the images are dirty and not clear.
  10. An in vivo model would be necessary to support their results, or, if beyond the scope, the manuscript’s results should be tuned down and the limitations acknowledged
  11. This reviewer personally misses key references regarding the topic: a project focusing on multiple myeloma pioneered initial observations that the expression level of JAM-A by malignant plasma cells can predict disease outcome (PMID: 29064484). Subsequently, elevated membrane expression of JAM-A also on bone marrow endothelial cells of patients with newly diagnosed or relapsed-refractory multiple myeloma cells predicted poor clinical outcome was also uncovered (PMID: 32354870). The authors should expand introduction and discussion accordingly.
  12. In the frame of this thinking, Solimando et al. uncovered that the expression of JAM-A on the surface of cancer-associated endothelial cells was inversely correlated with ADAM17 expression. Conversely,soluble-sJAM-A release correlated directly with ADAM17 upregulation, a mechanism described for endothelial cells in inflammation.ADAM17 upregulation has also been observed in MM in the context of fractalkine release, which identifies this system as a potential novel therapeutic target in MM patients to disrupt a vicious circle enhancing the tumor niche. This can be a significant dynamic and context dependent limitation for the authors statement and results (PMID: 32354870). Please substantiate.
  13. JAM-A is highly expressed on vessels wall and platelets (also within megakaryocytes). This would be a major concern in JAM-A targeting via CAR-T. Expand and discuss please
  14. Do the authors have any data showing how the binding domain of the CAR cell could

bind to one single JAM-A molecule simultaneously to achieve avidity or to other homo and heterotypic partners (i.e. CD9, LFA-1, etc ?

  1. How the binding happens at ultrastructural level?
  2. Given the tissue expression of JAM-A, regarding the CRS risk, the median time of this clinical picture is more likely to be precocious or delayed in a putative clinical setting and would the author expect a more severe scenario in case of earlier one? This can be related to the T cell expansion, therefore based on the expecteddynamics this can be at least partially predicted. Can the author comment on this topic?
  3. Would the authors see any potential correlation between the CRS grade and the cancer cellburden?
  4. Regarding the neurotoxicity, the authors mention this compelling topic. It would be therefore worth to uncover the potential mechanism that underly this undesired effect (again, can the high tumour burden represent a risk factor as well as CRS, the highexpansion and persistence of the CAR-T cells?
  5. Could the author comment about the potential use of their product for inpatients vs outpatient setting?
  6. While manufacturing, did the author take into account to use fresh or cryopreserved T cell products affectingthe CAR-T production efficiency?
  7. Can the author comment on the immunophenotype of the CAR-T after the manufacturing?
  8. In more details, have the author compared the product immunophenotype with the oneobtained with longer and more traditional and are there any differences in T cell differentiation in terms of CD4 and CD8 ratios?
  9. While dealing with transplant we usually deal with early on or delayed cell harvesting, followed by collection and storage. Would this strategy be suitable for the authors’model?
  10. Since the PFS (DFS) is still not optimal for cancer patients in the context of CAR-T study, is there any chance to design study comprising tandem strategies also in the context of this novel construct? I feel this hypothesis limitations given the uncertainty about re-treatement in the context of CAR-T strategies. Do the authors envision a given cancer scenarios?

Author Response

Thank you.

Reviewer 3 Report

The manuscript entitled:"Using the power of Junctional Adhesion Molecules combined with the target of CAR-T to inhibit cancer proliferation and metastasis, and eradicate tumors" focused on the evaluation of a new approach based on CAR-T adoption to reduce cancer proliferation is wel lwritten and requires minor revisions to be suitable for publication

  • In the text, the authors well report the technical aspects related to the adoption of this approach in pre clinical models. Please, could the authors also evaluate the role of this approach in clinical setting? How this approach may be usefull for the clinical setting?
  • In the conclusion section, few details about the most critical technical and biological aspects are discussed. Please, could the authors improve this section? 

Author Response

Thank you.

Round 2

Reviewer 2 Report

The authors have clarified some of the questions I raised in my previous review.

Unfortunately, most of the major problems have not been addressed by this revision (rebuttal took just a few days!); nonetheless, at least the author should improve and answer more precisely to the issue

  1. The image quality for the scratch is suboptimal and a robust quantification would be needed to make the results trustworthy and reproducible (There are open-source programs such as ImageJ to analyze images of in vitro scratch wound healing assays, but these tools require manual tuning of various parameters, which is time-consuming and limits image throughput. For that reason, an optimized plugin for ImageJ to automatically recognize the wound healing size, correct the average wound width by considering its inclination and quantify other important parameters such as area, wound area fraction, average wound width, and width deviation of the wound images obtained from a scratch/ wound healing assay has been developed. This would make the experiment consistent." Indeed, the authors' statement "The role of the Wound Healing assay is qualitative. This tool can be quantitative, but we reserved that role for the invasion experiments. We are aware of ImageJ and its properties, yet there is variability in the Wound Healing Assay, and using silicon inserts is difficult." This is partially true: a) the authors should insert a Scale bar; b) instruction for quantification and decreased approximation can be found at PMID: 25482647.

  2. Scale bare should also be applied to figure 6: these criteria are worth making the representative figure trustable and potentially reproducible.
  3. The p values on histograms in figure 5 should also be included.
  4. If the authors' feeling is that the above-mentioned observations are beyond the scope of this manuscript (as several additional points raised in my previous revision by the authors with the statement: "This is beyond the scope of our current manuscript") a clear explanation of the manuscript's limitation should be given to the audience rather than neglect the criticisms, in other to provide an unbiased and objective report of the data to the scientific community. This can be acceptable from the Editorial Office but would need clear and transparent statements.

Author Response

Dear Reviewer:

We appreciate your comments. We have worked on your suggestions as follows:

  1. The speed of the rebuttals is determined by the Editor. The first round gave us only 5 days to respond. In this second round we have 10 days to do so.
  2. The Wound Healing Assay. The company that supplies the silicone inserts (IBIDI) helped us to perform an analysis of the imaging obtained. Data analyzed using FastTrack AI (IBIDI, Gräfelfing, Germany). Now, Figure 3 contains both qualitative and quantitative data.
  3. Images with cells (Figure 3 and Figure 6) contain Scale bars.
  4. The p values on histograms in figure 5 are now included.

We appreciate greatly your efforts and ultimate help in strengthening our manuscript.

Sincerely,

Dario Mizrachi, Ph.D.

Round 3

Reviewer 2 Report

In this manuscript version the authors addressed some of the requests while deliberately making changes in another part of the manuscript that was previously improved based on the first review round. I quote:

"The role of JAM proteins in cancer is complex. JAM function by interacting with other proteins via several mechanisms: direct cell-cell interaction on adjacent cells, stabilization of adjacent cell surface receptors on the same cell, and interactions between JAM and cell surface receptors expressed on adjacent cells. The diverse interactions contribute to both the pro-and antitumorigenic functions of JAM[42]. This paradigm can be observed also in a study that presents evidence that JAM-A knockdown accelerates the proliferation and migration of human keratinocytes[10]. On the other hand, Solimando and colleagues studied the role of JAM-A in multiple myeloma (MM). In vitro JAM-A inhibition impaired MM migration, while in vivo treatment with an anti-JAM-A monoclonal antibody impaired tumour progression[43]. These results could correspond to JAM-A interactions and effects within the same cell or to a not well understood signal transduction[5,6]. The importance of mechanical transduction[44] from cellular junctions, both TJ and AJ, is poorly understood but the work by Solimando[43,45] and others[46,47] really highlight the need to further study of this phenomenon." The text is modified in another section with no apparent explanation. Indeed the current manuscript version contains only 47 references. The previously revised version 60 Instead.

This reviewer again misses data in the introduction and discussion regarding available research, patent analysis and existing platform for the proposed models, already available. This can be ameliorated: describe how our project goes beyond the state-of-the-art, and the extent to which the proposed work is ambitious. It was already pointed out that this reviewer personally missed key references regarding the topic: I quote the previous revision: "a project focusing on multiple myeloma pioneered initial observations that the expression level of JAM-A by malignant plasma cells can predict disease outcome (PMID: 29064484). Subsequently, elevated membrane expression of JAM-A also on bone marrow endothelial cells of patients with newly diagnosed or relapsed-refractory multiple myeloma cells predicted poor clinical outcome was also uncovered (PMID: 32354870)". The authors previously expanded the introduction and discussion accordingly. Now, this has been neglected and no highlights in yellow pointed this out. 

I quote the previous rebuttal"

  1. In the frame of this thinking, Solimando et al. uncovered that the expression of JAM-A on the surface of cancer-associated endothelial cells was inversely correlated with ADAM17 expression. Conversely,soluble-sJAM-A release correlated directly with ADAM17 upregulation, a mechanism described for endothelial cells in inflammation.ADAM17 upregulation has also been observed in MM in the context of fractalkine release, which identifies this system as a potential novel therapeutic target in MM patients to disrupt a vicious circle enhancing the tumor niche. This can be a significant dynamic and context dependent limitation for the authors statement and results (PMID: 32354870). Please substantiate.

The authors previously answered

"We have included a paragraph to highlight the work you have mentioned. Solimando’s observations include fully functional JAM-A, meaning capable of cis- and trans-interactions as well as signal transduction. Further understanding of the origin of the phenotypes is not clear in Solimando’s research as well as ours. We provide increased cell-cell interactions while Solimando’s approach is to prevent JAM-A interactions. Our biologic does not transduce signal while Solimando’s approach does. Little is known of such intracellular signaling. Further studies will address this difference. One final point, regarding the overexpression of JAMs in platelets, what would be the outcome of Solimando’s experiments on JAM-A expressed in platelets? There is much to learn very challenging, and very exciting."

Where are now these bullets? If the words limitation was one of the underlying reasons for this decision, I strongly recommend shortening and being concise in several other parts (perhaps moving in supplemental some details regarding already established techniques), giving merit to the time spent by both the reviewers and the authors to improve the manuscript scientific soundness previously. This also affects several other relevant manuscripts referenced in revision 1 and is absent from the current version.

In the author guidelines from this journal is written "Introduction: The introduction should briefly place the study in a broad context and highlight why it is important. It should define the purpose of the work and its significance, including specific hypotheses being tested. The current state of the research field should be reviewed carefully and key publications cited. Please highlight controversial and diverging hypotheses when necessary. Finally, briefly mention the main aim of the work and highlight the main conclusions. Keep the introduction comprehensible to scientists working outside the topic of the paper."

And in discussion guidelines "Authors should discuss the results and how they can be interpreted in perspective of previous studies and of the working hypotheses. The findings and their implications should be discussed in the broadest context possible and limitations of the work highlighted. Future research directions may also be mentioned. This section may be combined with Results"

In order to match these standards, I strongly recommend to reconsider the previous comments (accomplished by the authors in the first version), integrating with the improvement made in the current manuscript version.

Minor

What happened to Figure 6? Specifically, the image quality is still poor (blurry) the scale bars are still missing in the top panel, but now the panel is duplicated in a superimposed figure with a scale bar. I think is a mare copy paste replacement issue. Please fix

Author Response

please accept our apologies for the misunderstanding of downloading an incompatible version fo the revised manuscript.

CHanges have been made to the introduction and the discussion.

Figure 6 has been updated as per your request.

Dario Mizrachi

Round 4

Reviewer 2 Report

The authors have clarified the questions I raised in my previous reviews. Most of the major problems have been addressed by this revision. The manuscript's quality improved.